# Recessive *TMOD1* mutation causes childhood cardiomyopathy

Catalina Vasilescu[1,13], Mert Colpan [2,13], Tiina H. Ojala [3], Tuula Manninen[1], Aino Mutka [4], Kaisa Ylänen[5], Otto Rahkonen[3], Tuija Poutanen [5], Laura Martelius[6], Reena Kumari[7], Helena Hinterding[1], Virginia Brilhante[1], Simo Ojanen[1], Pekka Lappalainen [7], Juha Koskenvuo[8], Christopher J. Carroll[1,9], Velia M. Fowler[10], Carol C. Gregorio [2,11✉] & Anu Suomalainen [1,12✉]

Familial cardiomyopathy in pediatric stages is a poorly understood presentation of heart disease in children that is attributed to pathogenic mutations. Through exome sequencing, we report a homozygous variant in tropomodulin 1 (*TMOD1*; c.565C>T, p.R189W) in three individuals from two unrelated families with childhood-onset dilated and restrictive cardiomyopathy. To decipher the mechanism of pathogenicity of the R189W mutation in TMOD1, we utilized a wide array of methods, including protein analyses, biochemistry and cultured cardiomyocytes. Structural modeling revealed potential defects in the local folding of TMOD1[R189W] and its affinity for actin. Cardiomyocytes expressing GFP-TMOD1[R189W] demonstrated longer thin filaments than GFP-TMOD1[wt]-expressing cells, resulting in compromised filament length regulation. Furthermore, TMOD1[R189W] showed weakened activity in capping actin filament pointed ends, providing direct evidence for the variant's effect on actin filament length regulation. Our data indicate that the p.R189W variant in *TMOD1* has altered biochemical properties and reveals a unique mechanism for childhood-onset cardiomyopathy.

[1] Research Programs Unit, Stem Cells and Metabolism, Biomedicum-Helsinki, University of Helsinki, 00290 Helsinki, Finland. [2] Department of Cellular and Molecular Medicine and Sarver Molecular Cardiovascular Research Program, The University of Arizona, Tucson, AZ 85724, USA. [3] Department of Pediatric Cardiology, Helsinki University Hospital and University of Helsinki, 00290 Helsinki, Finland. [4] Department of Pathology, Helsinki University Hospital and University of Helsinki, 00290 Helsinki, Finland. [5] Tampere Center for Child, Adolescent and Maternal Health Research, Faculty of Medicine and Health Technology, Tampere University and University Hospital, 33521 Tampere, Finland. [6] Department of Pediatric Radiology, Helsinki University Hospital and University of Helsinki, 00290 Helsinki, Finland. [7] HiLIFE Institute of Biotechnology, University of Helsinki, 00014 Helsinki, Finland. [8] Blueprint Genetics, 00290 Helsinki, Finland. [9] Molecular and Clinical Sciences, St. George's, University of London, London, United Kingdom. [10] Department of Biological Sciences, University of Delaware, Newark, DE 19711, USA. [11] Cardiovascular Research Institute, Department of Medicine, Icahn School of Medicine, New York, NY 10029, USA. [12] HUSlab, Helsinki University Hospital, University of Helsinki, 00290 Helsinki, Finland. [13] These authors contributed equally: Catalina Vasilescu, Mert Colpan. ✉email: carol.gregorio@mssm.edu; anu.wartiovaara@helsinki.fi

Childhood cardiomyopathies are rare and severe heart muscle disorders characterized by pathological changes of the myocardium, unexplained by abnormal loading conditions or congenital heart disease[1]. A large proportion of childhood cardiomyopathies is considered to have a genetic origin, but the spectrum of causative genes is far from being completely elucidated. Many pathogenic variants affect the structure and function of the sarcomere, the contractile unit of the muscle, which has a highly organized protein machinery specialized in the generation of force. The sarcomere's primary structural proteins are myosin and actin, major components of the thick and thin filaments, respectively. Thick filament pathogenic variants in both alpha and beta-myosin heavy chain proteins, encoded by *MYH6* and *MYH7*, as well as their regulatory light chain proteins (*MYL2*, *MYL3*, *MYLK2*), have been associated with different types of cardiomyopathies. Thin filament pathogenic variants in alpha-actin (*ACTC1*), as well as in the associated tropomyosin 1 (*TPM1*) and troponin complex (*TNNC1*, *TNNT2*, *TNNI3*) are also well-known causes of cardiomyopathy[2]. Alpha-cardiac actin assembles into a polymerized strand called filamentous actin (F-actin) that provides a platform for troponin and tropomyosin binding, forming the thin filament in muscle cells. The actin filament presents a barbed end with rapid polymerization and a pointed end with slow polymerization. In cardiomyocytes, the barbed ends of sarcomeric F-actin are capped by CapZ and anchored to the Z-disc[3], while the pointed ends are capped by tropomodulin 1 (TMOD1) and located in the A-band[4]. TMOD1 is a member of the conserved TMOD family of proteins that cap F-actin pointed ends[5]. TMOD1 promotes actin filament stability by inhibiting actin monomer addition and dissociation and enhancing tropomyosin binding at the filament-pointed end. Leiomodins (*LMOD*), which also belong to the TMOD family of proteins, compete with TMODs for binding at the thin filament pointed ends and allow filament elongation to fine-tune thin filament lengths[6–8]. It is known that the cardiac muscle-specific isoform, *LMOD2*, is essential for heart development and function since the complete loss of Lmod2 in mice leads to cardiac death, and *LMOD2* variants are reported to cause severe neonatal cardiomyopathy in humans[9–15].

The knockout of *Tmod1* in mice was shown to be embryonic lethal due to arrested heart development from disrupted myofibril assembly[16]. However, unlike its counterpart LMOD2, TMOD1's importance in human health has remained unknown. This study reports the first homozygous *TMOD1* pathogenic variant to our knowledge, p.R189W, as a cause of dilated and restrictive cardiomyopathy in three pediatric patients from two unrelated families. Our investigation reveals compromised domain structure, loss of binding affinity to actin, and impairment in the ability to restrict thin filament lengths in cardiomyocytes by TMOD1 when harboring the R189W substitution. These findings reveal the mechanism behind the pathogenicity of the p.R189W variant in *TMOD1* and emphasize the role of TMOD1 as an important regulator of thin filament lengths and myofibril assembly in human cardiac muscle during childhood.

## Results

**Clinical features**. **Patient 1** (Family 1, Fig. 1a–c), a daughter of non-consanguineous healthy parents, was diagnosed at 12 years of age with dilated cardiomyopathy and short ventricular tachycardia episodes. Cardiac magnetic resonance (CMR) imaging showed that both the right and left atria were mildly dilated, indicating diastolic dysfunction. The right ventricular systolic function was normal. The left ventricle (LV) was dilated with a moderate decrease in systolic function (LV ejection fraction EF 38%). Mild late enhancement in the mid-myocardium of the intraventricular septum was observed as a marker of focal mid-myocardial fibrosis (Fig. 1 and Supplementary Videos 1 and 2). Normal findings were observed in metabolic examinations and in the histological analysis of the muscle sample. Ventricular tachycardia responded to orally administered beta-blocker. Heart failure was further treated with enalapril and diuretics. Cardiac function stabilized to the level of mild-to-moderate systolic dysfunction during 10 years follow-up: LV EF 34–38%, and maximal oxygen consumption $VO_2$ max of 19 mL/kg/min (57%) measured by spiroergometry (New York Heart Association classification, NYHA 1-2).

**Patient 2** (Family 2, Fig. 1d–f), a girl, is a daughter of non-consanguineous healthy parents. When she was 9 years old, she was referred to a pediatric cardiologist due to atrial enlargement on the 12-lead electrocardiogram, which was incidentally found as part of her workup for migraines. Transthoracic echocardiography revealed severe right and left atrial dilation with preserved left ventricular (LV) systolic function and normal wall thickness. A right ventricular systolic pressure of 46 mmHg was estimated as tricuspid regurgitation. CMR was done originally at the time of the diagnosis with normal ventricular volume and systolic function as well as myocardial findings. She was asymptomatic at the diagnosis time, with a $VO_2$ max of 35 mL/kg/min (85%) measured by spiroergometry.

Cardiac catheterization was performed soon after the diagnosis and demonstrated normal mean pulmonary artery pressure (21 mmHg) with an elevated left ventricular end-diastolic pressure of 17 mmHg. Her cardiac index was 2.7 l/min/m², with a normal pulmonary vascular resistance index of 1.6 Wu*m². At the age of 11 years, repeated cardiac catheterization was performed and demonstrated pulmonary arterial hypertension (mean pressure of 28 mmHg) with an elevated left ventricular end-diastolic pressure (19 mmHg) and increased pulmonary vascular resistance index at 3.8 Wu*m². At the time, the cardiac muscle biopsy sample showed no pathological findings.

At the age of 12.5 years, Patient 2 began to feel weak with dyspnea on effort (NYHA 2). CMR was repeated and demonstrated pericardial effusion and the dilatation of both atria accordingly as a marker of diastolic dysfunction. The right and left ventricular volumes were small. The right ventricular systolic function was normal, but the left ventricular systolic function was moderately decreased (LV EF 37 %). CMR delayed enhancement imaging showed no evidence of myocardial scar or focal fibrosis, but T1- mapping showed globally abnormal inversion times as a marker of diffuse myocardial fibrosis (Fig. 1, and Supplementary Videos 3 and 4).

A new spiroergometry measurement demonstrated a decreased $VO_2$ max of 24.5 mL/kg/min (51%). Warfarin therapy was started to prevent atrial thrombus. No conduction abnormalities or arrhythmias were detected during the follow-up. Patient 2 was listed for heart transplantation and received the transplant at the age of 13 years. Supplementary Fig. 1 shows the histological examination of diseased tissue at the time of heart transplant.

**Patient 3** (Family 2, Fig. 1g–i), the twin brother of Patient 2, was initially asymptomatic and underwent cardiac ultrasound screening at 10 years of age. His echocardiographic study revealed normal left ventricle dimensions with a mild decrease in systolic function, severe bi-atrial dilatation, and mild tricuspid regurgitation with a low calculated systolic gradient. CMR confirmed bi-atrial enlargement and normal ventricular end-diastolic volumes. Right ventricular systolic function was normal, but the left ventricular systolic function was mildly decreased (LV EF 45%). CMR delayed enhancement imaging showed no myocardial scar or focal fibrosis (Fig. 1 and Supplementary Videos 5 and 6); unfortunately, a T1-mapping sequence was not performed. Spiroergometry showed normal exercise capacity and $VO_2$ max of 43 mL/kg/min (100%). At the

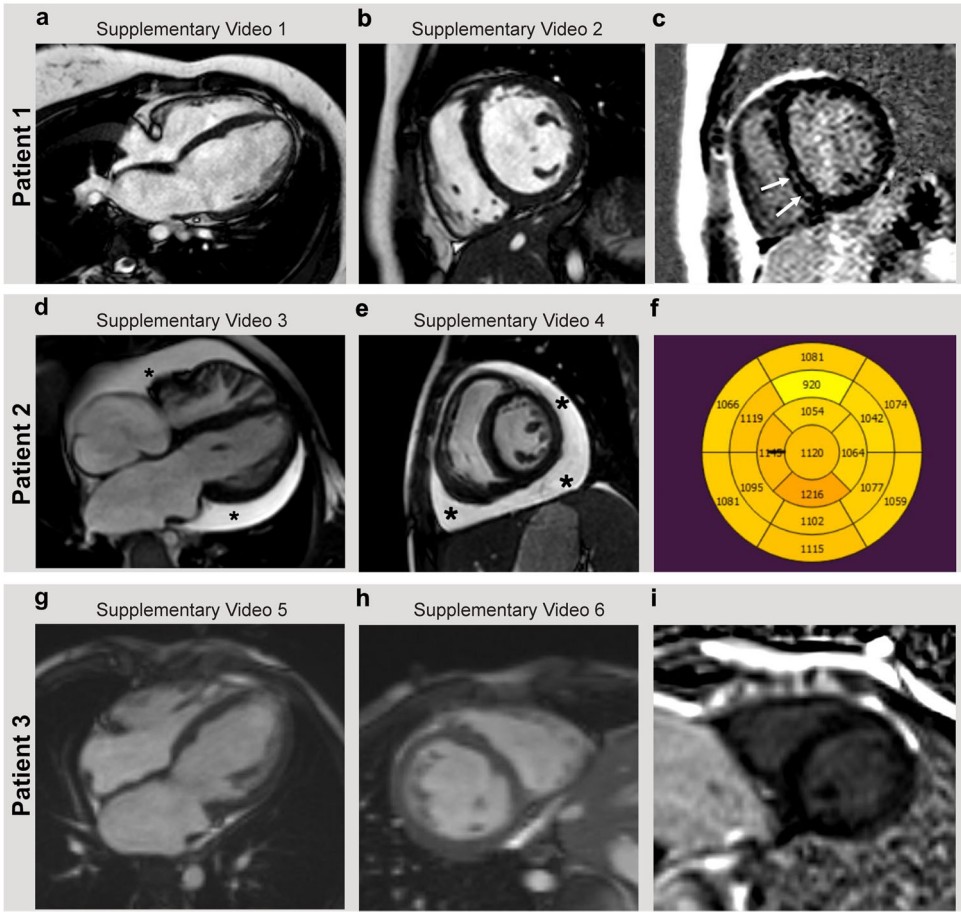

**Fig. 1 Cardiac magnetic resonance (CMR) imaging.** The linked videos, as indicated in each separate panel, are available as Supplementary Videos 1–6. **a**, **b** Patient 1 findings at the time of the diagnosis showed dilated cardiomyopathy (left ventricle end-diastolic volume 105 mL/m², Z+2.4, end-systolic volume 65 mL/m², Z+4.5) and poor left ventricular systolic function (ejection fraction EF 38%). The right ventricular size and the systolic function were normal (right ventricular end-diastolic volume 87 mL/m², Z-0.5 end-systolic volume 47 mL/m², Z+1, EF 46%) in four-chamber and short-axis cine. Both right and left atria were mildly dilated, as a marker of diastolic dysfunction. **c** Gadolinium-enhanced T1 fat-saturated images showed mild late enhancement in the mid-myocardium of the intraventricular septum (arrows). **d**, **e** Patient 2 findings at the time when she was listed for heart transplantation showed pericardial effusion (stars), small left and right ventricles, and bi-atrial dilatation (left ventricular end-diastolic volume 43 mL/m², Z-3 and left ventricular end-systolic volume 22 mL/m², Z-1.2). The left ventricular systolic function was moderately decreased (ejection fraction EF 37%). Delayed enhancement imaging showed no evidence of a myocardial scar or focal fibrosis. **f** However, T1 mapping showed globally abnormal inversion times as a marker of diffuse fibrosis. **g**, **h** Patient 3 demonstrated, at the time of the diagnosis, bi-atrial enlargement but normal ventricular end-diastolic volumes (left ventricular end-diastolic volume 75 mL/m², Z 0.5 and left ventricular end-systolic volume 41 mL/m², Z 2.7, right ventricular end-diastolic volume 55 mL/m², Z-2.9 and right ventricular end-systolic volume 28 mL/m², Z-0.9). The right ventricular systolic function was normal (EF 50%) but left ventricular systolic function was mildly decreased on CMR (EF 45%). **i** CMR delayed enhancement imaging showed no evidence of a myocardial scar. The T1 mapping sequence was not performed.

16-month follow-up, his echocardiographic study suggested pulmonary hypertension with an estimated right ventricular pressure of 50 mmHg. Subsequent cardiac catheterization at the age of 11.5 years demonstrated elevated pulmonary artery pressure (mean of 26 mmHg) with an elevated left ventricular end-diastolic pressure (22 mmHg). His cardiac index was 2.6 l/min/m² with a normal pulmonary vascular resistance index of 1.5 Wu*m².

After a 2-year follow-up period, at the age of 12 years, Patient 3 developed atrial tachycardia and non-sustained ventricular tachycardia. He became symptomatic with discomfort and upper-abdominal pain during exercise. Spiroergometry demonstrated a decreased VO₂ max of 33.2 mL/kg/min (64 %). He was started on beta-blocker and warfarin therapy, and he underwent ICD implantation for documented non-sustained ventricular tachycardia. He was listed for heart transplantation and received the transplant at the age of 12.5 years.

**Genetic findings.** Exome sequencing data analysis identified a homozygous missense variant in *TMOD1* (tropomodulin-1, NC_000009.11:g.100326397C>T (GRCh 37), NM_003275.3: c.565C>T, NP_003266:p.R189W) in all three patients (Fig. 2a, Supplementary Figs. 2 and 3). The parents are heterozygotes. The variant has a high CADD C-score of 33 and is rated deleterious by SIFT. No homozygotes existed in the gnomAD database (https://gnomad.broadinstitute.org/). In the gnomAD, the overall minor allele frequency is 0.00007, in Finland 0.00035, and in "other" populations 0.0004. These data suggest that the variant is not especially enriched in Finland but either arose independently on multiple haplotypes or is an old variant of very low frequency. The two Finnish families have no demonstrable or known common relatives and they come from different regions of the country. The three analyzed patients from Finland share a common haplotype of 3.3 Mb, which is short and proposes a common ancestry even 30 generations back.

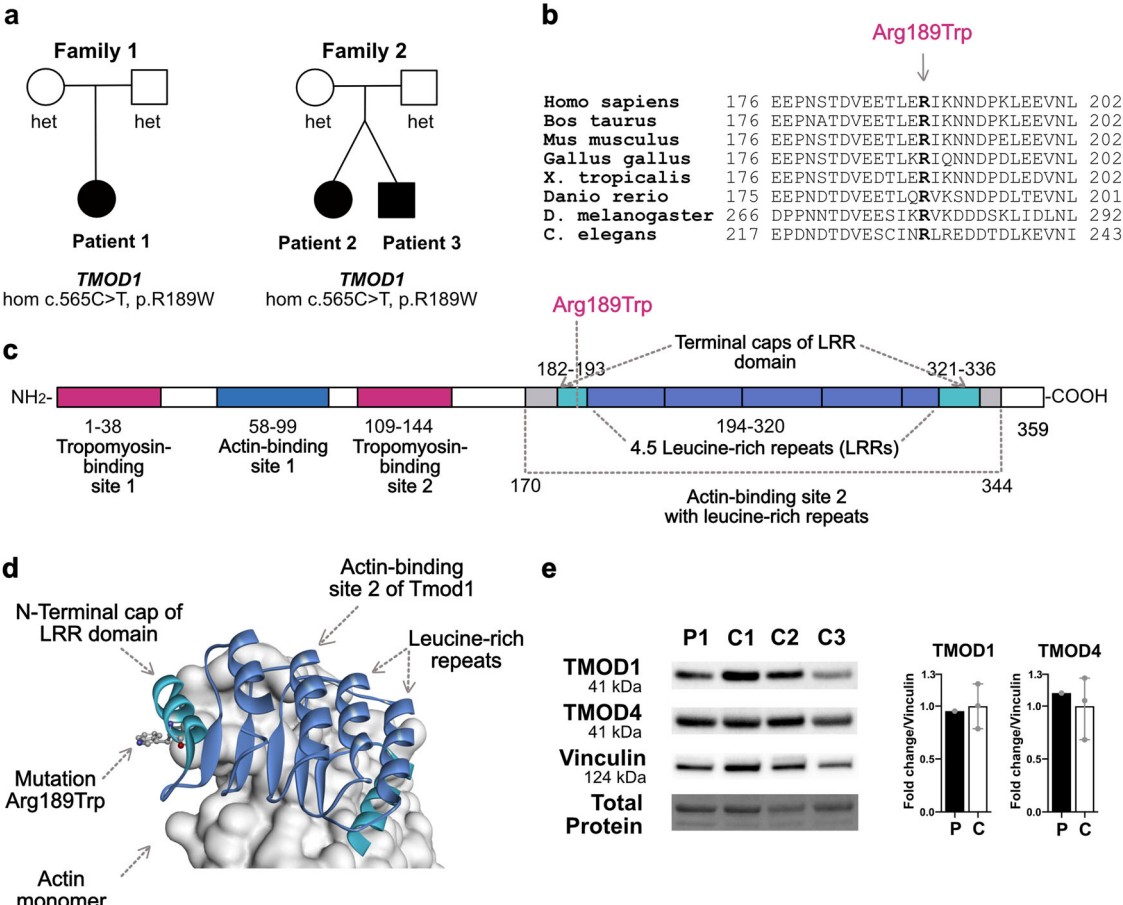

**Fig. 2 TMOD1 is identified as a disease gene for cardiomyopathy. a** Pedigrees of Family 1 and 2. **b** Multiple sequence alignment of TMOD1 homologs from different species. **c** Diagram of the protein with domains. **d** Modeling of patient variant R189W into the crystal structure of human actin-binding site 2 of TMOD1 (blue) in complex with an actin monomer (white) (Protein Data Bank ID: 4PKI). **e** Western blotting of TMOD1 and TMOD4 using protein lysates from skeletal muscle biopsies of Patient 1 and three controls. Quantification of TMOD1 and TMOD4 blots relative to vinculin. Data shown are mean ± standard deviation (*n* = 1 or 3 biologically independent samples). Patient 1 and control samples are shown in black and white bars, respectively.

These data indicate an ancient founder effect for the *TMOD1* p.R189W variant.

**Molecular modeling shows that the variant results in the exposure of a hydrophobic amino acid at the protein's surface.** The variant amino acid, Arg-189, is highly conserved across species (Fig. 2b). TMOD1 harbors two actin- and two tropomyosin-binding sites (Fig. 2c), enabling TMOD1 to cap the pointed end of tropomyosin-coated actin filaments and diminish the rate of addition and dissociation of actin subunits (see refs. [5,17] for reviews). The actin-binding sites 1 and 2 of human TMOD1 were separately crystallized bound to actin monomers[18]. Arg-189 localizes to actin-binding site 2 in the so-called "N-terminal cap" region of the leucine-rich region domain, which tightly contacts actin. The ascending loops of each of the four-and-a-half leucine-rich region motifs are in close contact with actin. The N- and C- terminal caps of the leucine-rich region domain, consisting of short alpha-helical regions, shield the domain's hydrophobic core from solvent exposure. Modeling of the p.R189W variant in the crystal structure of human actin-binding site 2 of TMOD1 in complex with an actin monomer (Protein Data Bank ID: 4PKI[18]) shows that the change results in the exposure of a hydrophobic amino acid at the domain's surface, with potential to influence the local folding of this domain, or its affinity for actin (Fig. 2d).

**Patient muscle biopsy showed normal levels of TMOD1.** TMOD1 is important for sarcomeric thin filament regulation both in cardiac and skeletal muscle, especially in slow fibers[19]. The availability of a skeletal muscle biopsy sample from Patient 1 enabled us to perform Western blot analysis of the skeletal muscle TMOD isoforms (TMOD1 and TMOD4). We found TMOD1 and TMOD4 levels in the skeletal muscle of Patient 1 to be similar to that of controls (Fig. 2e).

**TMOD1 localizes to thin filaments and sarcomere structure appears normal in induced pluripotent patient stem cell-derived cardiomyocytes.** We established a human-derived cardiomyocyte model to study the variant's effect in a relevant cellular context. We generated induced pluripotent stem cells (iPSCs) from patient and control fibroblasts and further differentiated them into cardiomyocytes. On day 20 of differentiation, the cells were purified to enrich the cardiomyocyte population and cultured further for maturation, including a 15-day culture in fatty acids and galactose medium. The cells were plated at day 45 for immunocytochemistry. The patient and control iPSC-derived cardiomyocytes were stained to reveal the sarcomeric structure and TMOD1 localization. Patient iPSC-cardiomyocytes developed a sarcomeric structure broadly similar to controls, with troponin-T flanking the Z line as expected for a thin filament protein (Fig. 3a), and TMOD1 located in the middle of the

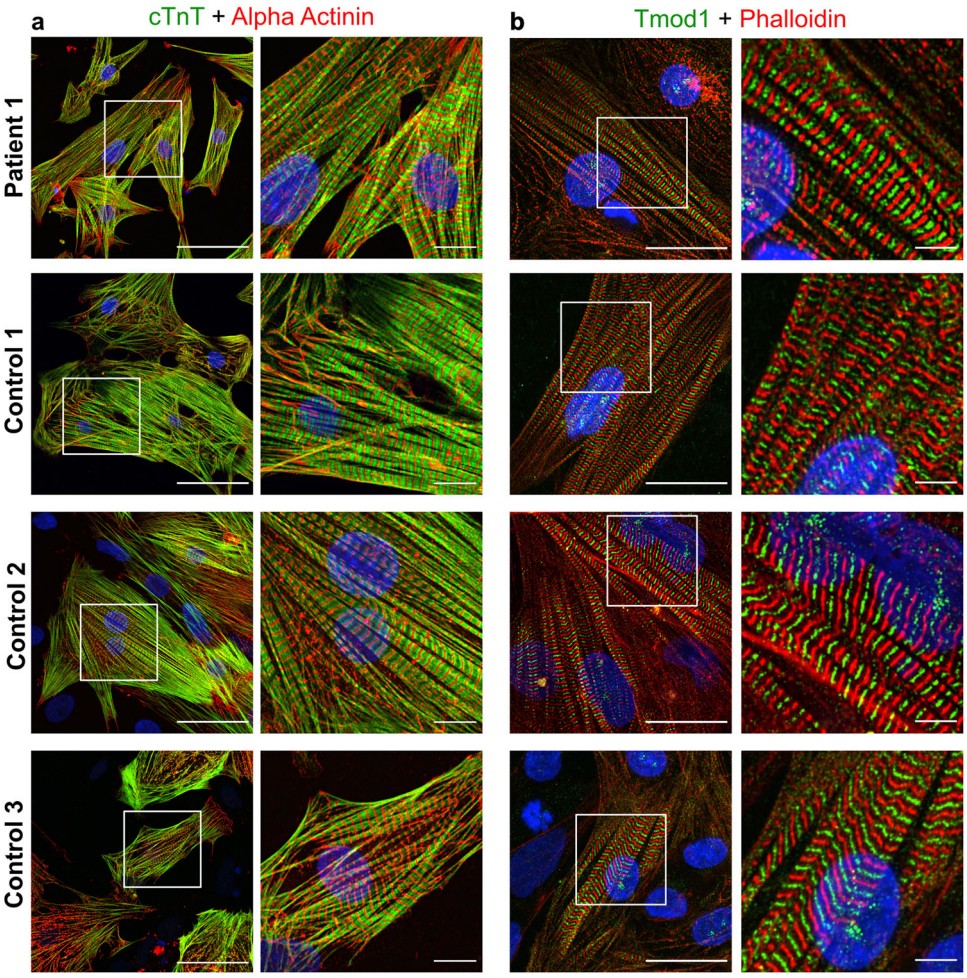

**Fig. 3 TMOD1 localizes correctly in patient cardiomyocytes. a** Sarcomeric structure in Patient 1 and three control iPSC-cardiomyocytes (Scale bars = 50 μm and 10 μm). cTnT: cardiac troponin T (green) and alpha-actinin (red): component of the Z-disk. **b** TMOD1 (green) localization in the patient and control iPSC-cardiomyocytes (Scale bars = 25 μm and 5 μm). Phalloidin (red) staining was used for visualization of filamentous actin. The regions in white boxes in left panels of (**a**) and (**b**) are shown at higher magnification in the right panels of (**a**) and (**b**). Note that the sporadic appearance of TMOD1 in both control and patient samples as a broader band (doublet) is linked to the precise relaxation of the sarcomere and/or the angle of the myofibril that is being photographed.

sarcomere (at thin filament pointed ends), as expected (Fig. 3b). TMOD1 appears as a single band rather than as a doublet (Fig. 3b), indicating that the sarcomeres were not fully relaxed in either the control or patient cells under the fixation conditions used here. The preferential Z-line staining by phalloidin rather than along the entire thin filament length is often observed in cardiac myocytes in which sarcomeres are not fully relaxed[20] and is not different between control and patient iPSC-derived cardiomyocytes (Fig. 3b). A fraction of TMOD1 is also present in the cytoplasm, and the cytoplasmic amount seems to be inversely related to the degree of sarcomeric development (Supplementary Fig. 4) in agreement with previous observations from embryonic chick cardiomyocytes[21]. These results suggest that TMOD1[R189W] localizes correctly to the thin filaments and does not disrupt sarcomere formation and structure, consistent with the late childhood disease onset.

**TMOD1[R189W] is a weaker cap of actin filament pointed ends compared to TMOD1[wt].** TMOD1 prevents the polymerization of actin filaments by capping their pointed ends[22]. To test if the p.R189W variant affects TMOD1's ability to inhibit actin pointed end polymerization, we performed pyrene-actin polymerization

assays in the presence of purified recombinant human TMOD1. For experiments in the absence of TPM1, we performed two independent assays using short (Fig. 4a) or long actin filaments (Supplementary Fig. 5A) capped at their barbed ends by gelsolin (1:10 or 1:100 gelsolin-to-actin ratio, respectively). Short filaments were generated to minimize the spontaneous fragmentation of long filaments and subsequent polymerization from the newly formed barbed ends, allowing for accurate measurements of TMOD1's pointed end-capping activity. Actin polymerization rates in the absence (control) or presence of various concentrations of TMOD1[wt] or TMOD1[R189W] (Fig. 4b and Supplementary Fig. 5B) were extracted from pyrene-actin fluorescence curves to calculate their capping affinities (Supplementary Tables 1 and 2). Fitting the data to an exponential decay model revealed the capping activity of TMOD1[R189W] to be ~2-fold weaker than that of TMOD1[wt] at either filament length. In the presence of TPM1, TMOD1[wt] and TMOD1[R189W] capped filament pointed ends more effectively (Fig. 4c), but TMOD1[R189W] still functioned as a weaker cap compared to TMOD1[wt] (Fig. 4d and Supplementary Table 1).

In addition to slowing actin polymerization at the pointed ends, TMOD1 is also able to inhibit filament depolymerization from the same end via its capping ability[22]. We utilized pyrene-actin

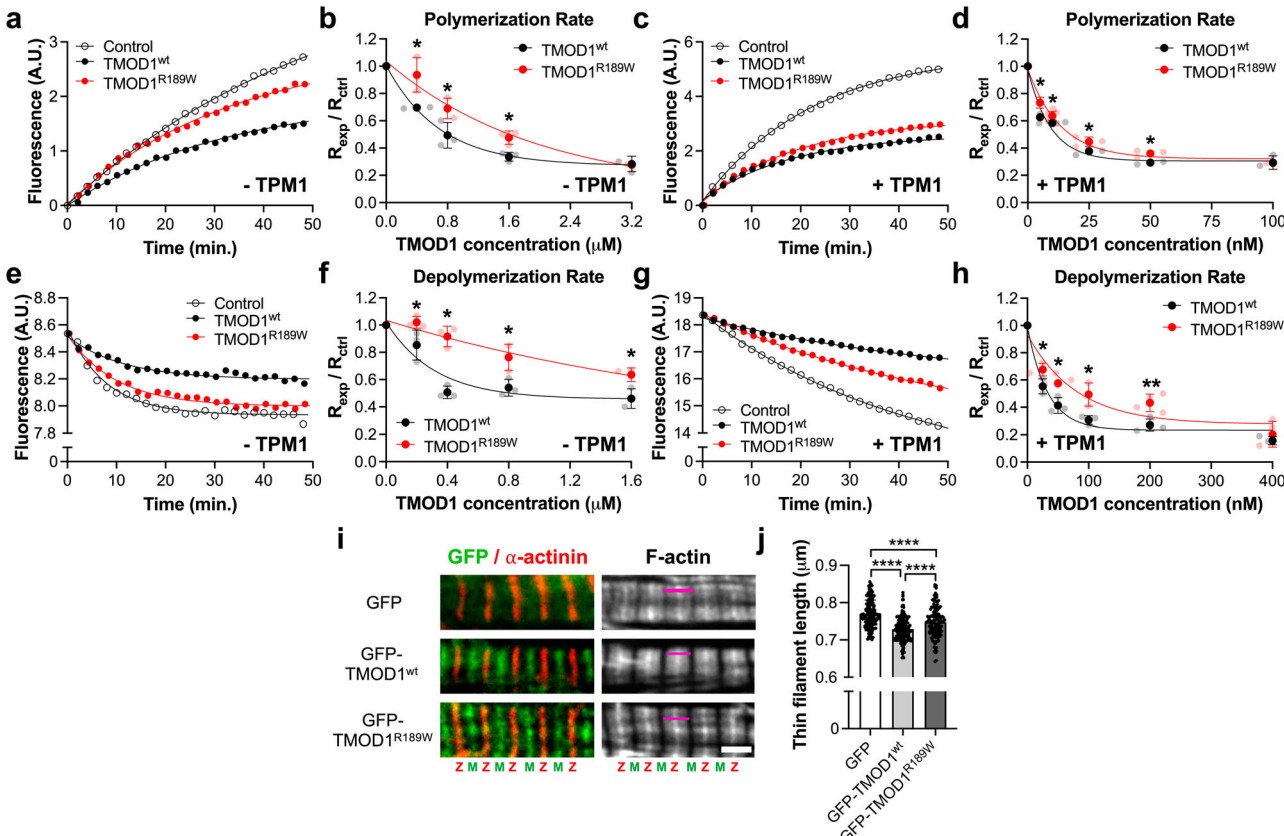

**Fig. 4 The p.R189W variant weakens TMOD1's ability to cap the pointed ends of actin filaments. a** Representative curves of pyrene-actin fluorescence (A.U.: arbitrary units) from pointed ends of gelsolin-capped actin filaments (1:10 gelsolin-to-actin molar ratio) over time (min) in the presence of 0.8 μM TMOD1$^{wt}$ or TMOD1$^{R189W}$ (control: spontaneous actin pointed end polymerization) and in the absence of TPM1 (−TPM1). **b** Concentration-dependent inhibition of actin polymerization in the presence of TMOD1$^{wt}$ or TMOD1$^{R189W}$ ($p$-value = 0.0299, 0.0480, 0.0146, 0.8498, respectively). **c** Representative curves of pyrene-actin fluorescence (A.U.) from pointed ends of gelsolin-capped actin filaments (1:100 gelsolin-to-actin molar ratio) over time (min) in the presence of TPM1 (+TPM1), with 25 nM TMOD1$^{wt}$ or TMOD1$^{R189W}$. **d** Concentration-dependent inhibition of actin polymerization in the presence of TPM1 and TMOD1$^{wt}$ or TMOD1$^{R189W}$ ($p$-value = 0.0153, 0.0474, 0.0485, 0.0161, 0.9125, respectively). **e** Representative curves of decrease in pyrene-actin fluorescence (A.U.) resulting from depolymerization from the pointed ends of gelsolin-capped actin filaments (1:10 gelsolin-to-actin molar ratio) over time (min) in the presence of 0.4 μM TMOD1$^{wt}$ or TMOD1$^{R189W}$ (control: spontaneous actin pointed end depolymerization), in the absence of TPM1 (−TPM1). **f** Concentration-dependent inhibition of actin depolymerization in the presence of TMOD1$^{wt}$ or TMOD1$^{R189W}$ ($p$-value = 0.0329, 0.0165, 0.0279, 0.0125, respectively). **g** Representative curves of decrease in pyrene-actin fluorescence (A.U.) resulting from depolymerization from the pointed ends of gelsolin-capped actin filaments (1:100 gelsolin-to-actin molar ratio) over time (min) in the presence of TPM1 (+TPM1), with 100 nM TMOD1$^{wt}$ or TMOD1$^{R189W}$. **h** Concentration-dependent inhibition of actin depolymerization in the presence of TPM1 (+TPM1) and TMOD1$^{wt}$ or TMOD1$^{R189W}$ ($p$-value = 0.0411, 0.0104, 0.0237, 0.0056, 0.4570, respectively). Actin polymerization or depolymerization rates relative to the control ($R_{exp}/R_{control}$) were calculated as the first derivatives at time zero after exponential fit. Data shown are mean ± standard deviation ($n = 3$ independent experiments, *$p < 0.05$, **$p < 0.01$, Student's $t$-test). Control, TMOD1$^{wt}$ and TMOD1$^{R189W}$ are shown in white, black and red circles, respectively. **i** Immunofluorescence images of neonatal rat cardiomyocytes fixed and stained with Texas Red-conjugated phalloidin to mark F-actin and an anti-α-actinin antibody to indicate the Z-disc ("Z") after being transfected with GFP (control), GFP-TMOD1$^{wt}$ or GFP-TMOD1$^{R189W}$ (Scale bar = 2 μm). Magenta bars show representative thin filament arrays (two bundles of thin filaments extending into opposite half-sarcomeres toward the M line ["M"]) that length measurements were taken from. **j** Thin filament length measurements collected from F-actin images of transfected cells (GFP = 0.770 ± 0.003 μm; GFP-TMOD1$^{wt}$ = 0.729 ± 0.003 μm; GFP-TMOD1$^{R189W}$ = 0.748 ± 0.003 μm). Data shown are mean ± standard error of the mean ($n = 3$ independent cultures, 11 cells per culture, 5 measurements per cell, 3 sarcomeres per measurement for a total of 141, 162 or 133 independent measurements, ****$p < 0.0001$, One-way ANOVA). GFP, GFP-TMOD1$^{wt}$ and GFP-TMOD1$^{R189W}$ are shown in white, light gray and dark gray bars, respectively.

depolymerization assays to test if the p.R189W variant also affects TMOD1's ability to prevent spontaneous actin depolymerization from actin filament pointed ends. Similar to the polymerization assays, we performed this experiment using pyrene-labeled short (Fig. 4e) or long actin filaments (Supplementary Fig. 5C) capped at their barbed ends by gelsolin (1:10 or 1:100 gelsolin-to-actin ratio, respectively) in the absence of TPM1. Actin depolymerization rates in the absence (control) or presence of various concentrations of TMOD1$^{wt}$ or TMOD1$^{R189W}$ (Fig. 4f, Supplementary Fig. 5D) were extracted from pyrene-actin fluorescence curves to calculate their

capping affinities (Supplementary Tables 1 and 2). Inhibition of actin pointed end depolymerization by TMOD1$^{R189W}$ was weaker by ~5-fold or ~2-fold than TMOD1$^{wt}$ at shorter and longer filament lengths, respectively.

In the presence of TPM1, TMOD1$^{wt}$ and TMOD1$^{R189W}$ inhibited depolymerization from filament pointed ends more effectively (Fig. 4g), but capping by TMOD1$^{R189W}$ was still significantly weaker than that by TMOD1$^{wt}$ (Fig. 4h and Supplementary Table 1). In summary, we found that the p.R189W variant weakens the ability of TMOD1 to inhibit

spontaneous actin polymerization and depolymerization from actin filament pointed ends in the absence or presence of TPM1.

To test the effect of the p.R189W variant on TMOD1's capping ability via an independent assay, we performed sedimentation assays for F-actin depolymerization in the presence of various concentrations of TMOD1[wt] or TMOD1[R189W]. Upon incubation with varying concentrations of TMOD1[wt] or TMOD1[R189], F-actin was depolymerized with latrunculin A (Lat A), a drug that rapidly disassembles F-actin by sequestering monomeric globular actin (G-actin). The remaining F-actin and G-actin after Lat A treatment were collected as pellets and supernatants, respectively, by ultracentrifugation and subjected to sodium dodecyl sulfate–polyacrylamide gel electrophoresis to determine the fraction of F-actin protected against depolymerization by TMOD1[wt] or TMOD1[R189W] (Supplementary Fig. 6A, C). Fitting the data from pellet or supernatant samples to a nonlinear dose-response model (Supplementary Fig. 6B, D) revealed that TMOD1[wt] inhibited rapid filament depolymerization by Lat A more effectively than TMOD1[R189W] (Supplementary Table 3). Therefore, the p.R189W variant impairs the ability of TMOD1 to inhibit actin depolymerization at the pointed ends. In summary, both assays (fluorescence and sedimentation) consistently demonstrated a reduction in the ability of TMOD1 to cap the pointed ends of actin filaments in the presence of the p.R189W variant. Therefore, the p.R189W variant significantly weakens the ability of TMOD1 to inhibit actin polymerization and depolymerization from actin filament pointed ends.

**GFP-TMOD1[R189W]-expressing rat cardiomyocytes demonstrate compromised thin filament length regulation.** Expression of excess levels of TMOD1 in cardiomyocytes shortens thin filament lengths by reducing actin association at pointed ends[23]. To test if the p.R189W mutation affects TMOD1's cellular function, we expressed GFP-TMOD1[wt], GFP-TMOD1[R189W], or GFP alone in neonatal rat cardiomyocytes. The p.R189W mutation did not disrupt the subcellular assembly of TMOD1 since GFP-TMOD1[wt] and GFP-TMOD1[R189W] localized to the pointed ends of thin filaments similarly (Fig. 4i). Thin filaments were stained using fluorescently labeled phalloidin, and their lengths were calculated from the pointed end (near the M-lines) to the barbed end (near the Z-line) measured by line scans (Fig. 4i, magenta bars). As expected from previous work, thin filaments in cardiomyocytes expressing GFP-TMOD1[wt] were shorter in length [$0.729 \pm 0.003\ \mu m$] compared to control GFP alone [$0.770 \pm 0.003\ \mu m$] (Fig. 4j). However, GFP-TMOD1[R189W] was unable to shorten the thin filaments to the same extent [$0.748 \pm 0.003\ \mu m$] as GFP-TMOD1[wt]. This result indicates that the p.R189W variant weakens TMOD1's ability to shorten thin filament lengths in cardiomyocytes.

## Discussion

Our evidence, including genetics, protein modeling, cultured cardiomyocytes, and biochemical data, indicate that to the best of our knowledge, *TMOD1* is a novel disease gene underlying childhood-onset dilated and restrictive cardiomyopathy. The findings of the same rare variant in two non-consanguineous families with a similar clinical manifestation, the variant's location in a highly conserved amino acid, and the lack of homozygotes in healthy populations strongly support the pathogenic role of the homozygous p.R189W substitution in TMOD1. While thin-filament gene defects are established causes of cardiomyopathies and nemaline myopathies[9,11–13,24–26], *TMOD1*, encoding a protein that localizes to the tip of the thin filament, is a new addition to this disease gene group.

In cardiomyocytes, TMOD1 functions in the context of the sarcomere. Many other sarcomeric proteins have been previously linked to cardiomyopathies, including TPM1, an interacting partner of TMOD1. The p.K15N variant in TPM1, causing familial dilated cardiomyopathy[27], negatively affects the binding properties of TMOD1 and LMOD2 to the F-actin pointed end[7], highlighting the close relationship between thin filament proteins in cardiomyocytes and linking TMOD1 variants to a similar pathological mechanism. Furthermore, variants in LMOD2, the functional counterpart of TMOD1, were reported to cause severe neonatal cardiomyopathy in humans[12,13,15], which highlights the importance of the TMOD family of proteins in cardiac health and function. Recently, the human pathogenic p.R1243H mutation in the *FLII (Flightless-I homolog)* gene, predisposing to cardiomyopathy, was investigated in a series of mouse models. These murine studies suggest that the *FLII* gene may regulate actin thin filament length by interacting with TMOD1 and preventing it from capping pointed ends[28].

TMOD1 is expressed in mammalian striated and smooth muscles, red blood cells, ocular lens fiber cells, and neurons. TMOD2 is neuronal-specific, TMOD3 is widely expressed, while TMOD4 is restricted to skeletal muscle[5]. Why the different TMODs or the related LMODs do not compensate for each other in the heart is unclear, as such synergy has been shown to occur in skeletal muscle[29]. TMOD1 is crucial for thin filament capping in cardiac muscle. *Tmod1*-knockout mice present with several abnormalities related to fetal heart development, vitelline circulation, and hematopoiesis, with lethality at approximately embryonic day 10[30,31]. Heart-specific Tmod1 expression rescued the lethality and indicated that the primary cause of the knockout death was the dysfunction of the myocardium[16]. This highlights the essential role of TMOD1 in the heart. More specifically, in the absence of Tmod1, the cardiomyocyte myofibrils did not become striated, and gaps in F-actin staining were not observed, which is indicative of impaired thin filament length regulation[31]. These murine data suggest that the isoform dynamics and compensation described for skeletal muscle do not apply to the same extent in the heart. This is supported by our findings, indicating that partially dysfunctional TMOD1 causes childhood-onset cardiomyopathy without skeletal myopathy. Furthermore, the TMOD1[R189W] variant localized correctly in the patient's stem cell-derived cardiomyocytes, not disrupting the sarcomere structure. This likely reflects the actual physiological situation, with the molecular and structural changes developing over time in the heart under contractile load and leading to manifestation in late childhood. The atrial dilatation that is present in the patients with the TMOD1 p.R189W missense variant is a consequence of poor ventricular function. In restrictive cardiomyopathy, the ventricular walls of the heart are rigid and cannot expand to fill up with blood sufficiently. This leads secondarily to an increase in atrial filling and dilatation. Therefore, restrictive cardiomyopathy is primarily a disorder affecting the ventricular muscle and the increased atria size reflects the degree of diastolic dysfunction in the ventricles.

We found TMOD1[R189W] to be capable of localizing to thin filament pointed ends. TMOD1 binding to the N-terminus of TPM1 is crucial for its subcellular assembly to the pointed end of the thin filament[32]. The p.R189W missense mutation is located downstream of the two TPM-binding sites of TMOD1; therefore, no direct effect on the interaction with TPM1 or on TMOD1 targeting to thin filament ends was expected. Indeed, we found TMOD1[R189W] to localize at the pointed ends of the thin filament successfully, but its ability to inhibit actin polymerization and depolymerization is significantly diminished on both biochemical and cellular levels. The leucine-rich region domain, where residue Arg-189 is located, contains the second actin-binding site of TMOD1. In agreement with our structural modeling, we found p.R189W to disrupt the interaction between TMOD1 and actin. As a result, TMOD1 capping of the pointed ends is weakened, challenging its ability to

maintain thin filament lengths. Similarly, pathological alterations in thin filament lengths have been found to underlie a number of human skeletal and cardiac myopathies[12,13,24–26]. These pieces of evidence strongly suggest that the molecular mechanism underlying TMOD1 cardiomyopathy caused by the p.R189W variant is the dysregulation of thin filament lengths.

## Methods

**Research subjects**. Genomic DNA from the patients and family members was extracted from peripheral blood using standard protocols. Primary fibroblasts were grown from skin biopsy samples using standard procedures. According to the Declaration of Helsinki, all participants gave their written consent for the study, and the ethics committee of Helsinki and Uusimaa Region Hospital approved the study plan. The cardiomyopathy diagnosis followed the statement of the European Society Working Group on myocardial and pericardial diseases. The sex of human patients is provided in the manuscript. The race or ethnicity of the patients is not provided to preserve their privacy under the European Data Protection Regulation. This manuscript did not study a population of subjects; only three patients from two unrelated families with the same homozygous gene mutation are reported. The patients were recruited due to hospitalization from dilated and restrictive cardiomyopathy.

**Exome sequencing**. Patient 1 was first investigated with a targeted-sequencing panel (Haloplex custom panel) of known cardiomyopathy genes, with no genetic findings. This prompted us to sequence the exome in a further quest to uncover a possible novel disease gene to the best of our knowledge. The exome data were filtered first with an assumption of a recessive inheritance model, resulting in the candidate homozygous variant. Co-segregation of the candidate variant was tested in the family members by Sanger sequencing.

Patients 2 and 3 were referred by clinicians to Blueprint Genetics for molecular diagnosis. Patient 2 was first investigated with a targeted gene panel, with negative findings. Both siblings and the parents were further tested by exome sequencing, identifying the same variant as in Patient 1. The haplotypes were defined by observation of common homozygous single-nucleotide variants along the chromosomal region of TMOD1.

**Cloning of TMOD1^wt and TMOD1^R189W**. For expressing GFP-TMOD1 in neonatal rat cardiomyocytes, mouse *Tmod1* cDNA was generated from mouse hearts and cloned into the pEGFP-C1 vector (Clontech, Mountain View, CA) using the XhoI and EcoRI restriction sites. The p.R189W mutation was introduced to pEGFP-C1-mTMOD1 via site-directed mutagenesis[33] using the following primer sequences: forward: 5′-A GAG GAA ACG CTG GAG TGG ATA AAG AAC AAT GAC CCA GAA CTA GAA GAG GT-3′ and reverse: 5′-CTT TAT CCA CTC CAG CGT TTC CTC TAC GTC TGT TGA ATT TGG TTC TTC ATC AG-3′. Substituted triplets are underlined.

For biochemical actin capping assays, human TMOD1 in pDONR221 plasmid (HsCD00042010) was obtained from DNASU Plasmid Repository[34]. GSN-bio-His plasmid encoding human plasma gelsolin was a gift from Gavin Wright (Addgene plasmid #52067; http://n2t.net/addgene:52067; RRID: Addgene_52067)[35]. Rat striated muscle α-tropomyosin (TPM1) in pET11d expression plasmid was a gift from Dr. Alla Kostyukova (Washington State University, Pullman, WA). For recombinant protein purification, TMOD1 with an N-terminal 6xHistidine-tag and gelsolin with a C-terminal 6xHistidine-tag were cloned into pReceiver-B01 (GeneCopoeia, Rockville, MD) using the KpnI and XhoI restriction sites. The p.R189W mutation was introduced to pReceiver-B01-TMOD1 via site-directed mutagenesis[33] using the following primer

sequences: forward: 5′- AG GAA ACG CTG GAA TGG ATA AAG AAC AAC GAC CCA AAA CTT GAA GAA GTT A -3′ and reverse: 5′- CTT TAT CCA TTC CAG CGT TTC CTC TAC GTC TGT TGA ATT TGG TTC TTC GT -3′. Substituted triplets are underlined. *DpnI* (New England Biolabs)-treated PCR products were transformed into DH5α *E. coli* cells (Life Technologies, Carlsbad, CA). Clones for pEGFP-C1 and pReceiver-B01 were selected by kanamycin and ampicillin resistance, respectively. Plasmids were purified using the ZR Plasmid Miniprep Classic kit (Zymo Research, Irvine, CA) according to the manufacturer's recommendations. Mutagenesis was confirmed with sequencing performed by Eton Bioscience (San Diego, CA). Oligonucleotide primers were synthesized by Sigma-Aldrich (St. Louis, MO).

**Protein expression and purification**. Actin acetone powder[36] obtained from rabbit skeletal muscle[37] was used to extract and purify G-actin. Recombinant TPM1 with an N-terminal Ala-Ser extension that mimics acetylation was purified as described previously[38]. His-tagged TMOD1 or gelsolin constructs were transformed into Rosetta™ 2 (DE3) pLysS competent cells (EMD Millipore) and grown to OD 0.6 in LB medium at 37 °C, 250 revolutions per minute (RPM). Protein expression was then induced with 0.1 mM isopropyl β-D-1-thiogalactopyranoside for 5 h, at 37 °C for gelsolin and room temperature for TMOD1. Cells were spun down at $5500 \times g$, 4 °C for 20 min. Pellets were resuspended in B-PER Bacterial Protein Extraction Reagent with DNase I, 1x Halt protease inhibitor cocktail (all from Thermo Fisher) and 0.1 mg/mL lysozyme (Sigma-Aldrich). After 20 min incubation at room temperature followed by 20 min of incubation on ice, the cells were sonicated and the extract was clarified by centrifugation at 16,000 RPM (Beckman-Coulter, JA-17 rotor), 30 min, 4 °C. The supernatant was transferred to the Ni-NTA Superflow column (Qiagen, Valencia, CA) after equilibrating with 50 mM sodium phosphate, pH 8.0, 300 mM sodium chloride, 10 mM imidazole (column buffer). The column was washed with column buffer extensively and His-TMOD1 or gelsolin-His was eluted with column buffer containing 250 mM imidazole. His-TMOD1 was dialyzed into 20 mM tris(hydroxymethyl)aminomethane-hydrochloric acid, pH 8.0, 150 mM sodium chloride, 1 mM ethylenediaminetetraacetic acid and gelsolin-His was dialyzed into 10 mM tris(hydroxymethyl)aminomethane-hydrochloric acid, pH 7.5, 150 mM sodium chloride, 0.1 mM magnesium chloride. Purified proteins were concentrated using Vivaspin® 15 R centrifugal concentrators (Fisher Scientific) and clarified by ultracentrifugation at 100,000 RPM (Beckman-Coulter TLA 120.2 rotor), 4 °C for 1 h. Protein concentrations were determined by Pierce™ BCA Protein Assay Kit (Thermo Fisher).

**Structural modeling**. We modeled the consequences of the TMOD1 homozygous variant using the software Discovery Studio 4.5 (Biovia). The actin-binding sites 1 and 2 of human TMOD1 were separately crystallized in complex with actin monomers due to competitive binding[18]. As a template for our model, we used the crystal structure of actin-binding site 2 (where the patient variant resides) in complex with an actin monomer (Protein Data Bank ID: 4PKI). The program generated five models, which were very similar to each other. The model with the lowest energy was chosen for further analysis. For clarity, the actin monomer was displayed as surface, and the protein motifs of actin-binding site 2 were emphasized by color. The variant site was displayed as "ball and stick".

**Cell culture**. We used several cell lines (neonatal rat cardiomyocytes, mouse embryonic fibroblasts, iPSCs derived from human primary fibroblasts and iPSCs-cardiomyocytes), which have

different nutrient requirements, and the different culture media are detailed below. Primary fibroblast cells were cultured in Dulbecco's modified Eagle's Medium-high glucose (Sigma, D6546-500ML) with 10% fetal bovine serum (FBS) (Sigma-Aldrich, F9665) and 1x GlutaMAX$^{TM}$-I (Gibco, 35050-038), supplemented with 1x penicillin/streptomycin (Gibco, 15070-063).

Human embryonic stem cell medium for iPSC cells contained Dulbecco's modified Eagle's Medium/F12 with Glutamine (Life Technologies, 31331-028), 20% KnockOut Serum Replacement (Gibco, 10828-010), 0.1 mM 2-mercaptoethanol (Sigma-Aldrich, M3148), 1x non-essential amino-acids (Gibco, 1110-035), 6 µg/mL recombinant human fibroblast growth factor basic protein (Peprotech, 100-18B), 0.25 mM sodium butyrate (MilliporeSigma$^{TM}$ TR1008G, Thermo Fisher Scientific). Induced pluripotent stem cells were grown in Essential 8 (E8) medium, which contained E8$^{TM}$ Basal Medium (Gibco, A15169-01) and 1x E8$^{TM}$ Supplement (Gibco, A15171-01).

iPSC-cardiomyocytes were derived and matured using cardiomyocyte glucose medium and cardiomyocyte fatty acid medium. Cardiomyocyte glucose medium (CDM3 glucose) consists of Roswell Park Memorial Institute Medium 1640 with L-glutamine and glucose (Gibco, 21875-034) supplemented with 500 µg/mL human serum albumin (Sigma-Aldrich, A9731-1G) and 213 µg/mL L-ascorbic acid 2-phosphate sesquimagnesium salt hydrate (Sigma-Aldrich, A8960-5G). Cardiomyocyte galactose and fatty acid medium (CDM3 galactose and fatty acids) contains Roswell Park Memorial Institute Medium 1640 with L-glutamine but without glucose (Gibco, 11879-020) supplemented with 500 µg/mL human serum albumin (Sigma-Aldrich, A9731-1G), 213 µg/mL L-ascorbic acid 2-phosphate sesquimagnesium salt hydrate (Sigma-Aldrich, A8960-5G), 0.025% glucose (Gibco, A2494001), 7.5 mM D-galactose (Sigma-Aldrich, G5388-100G), 100 µM oleic acid-albumin (Sigma-Aldrich, 03008-5 ML), 50 µM palmitic acid (Sigma-Aldrich, P0500-10G) conjugated with albumin, and 100 µM L-carnitine (Sigma-Aldrich, C0158-1G).

Rat cardiomyocytes were maintained in Dulbecco's modified Eagle's Medium with 1 g/L glucose (Gibco), 10% (vol/vol) FBS (HyClone) and 1% penicillin/streptomycin (Cellgro).

This study did not involve wild animals or field-collected samples. Neonatal mixed gender postnatal day 0–3 Sprague-Dawley rats or mixed gender mouse embryos were used as source for cell culture studies. None of the cell lines used were authenticated and no misidentified cell lines were used. The primary fibroblasts, iPSCs and iPSCs-cardiomyocytes were tested against mycoplasma contamination.

**Generation and characterization of induced pluripotent stem cells**. Low-passaged dermal primary fibroblasts were reprogrammed to induce pluripotent stem cells (iPSCs) from patient and control lines by neon electroporation system, transducing 700,000 cells with three plasmids containing Yamanaka factors (Oct3/4, Sox2, Klf4, and c-Myc): pCXLE-hOCT3/4-shp53-F, pCXLE-hSK, and pCXLE-hMLN (1 µg each). The settings for electroporation were 1650V, 10 ms, and 3 pulses. After electroporation, the cells were transferred into 6-cm freshly gelatinized dishes (0.15% gelatin) containing warm fibroblast medium (see "Cell culture" section for media recipes) on day 1. The media was changed on days 2 and 4. On day 5, feeder plates were prepared for passing the cells by seeding about 20,000 inactivated mouse embryonic fibroblasts (established from mouse embryos via standard procedures) per freshly gelatinized 6-cm dishes and culturing them overnight in fibroblast medium. On day 6, the electroporated cells were collected with TrypLE$^{TM}$ Express (Gibco, 12604-013), pelleted by centrifugation, resuspended in fibroblast medium, and seeded 150,000–300,000 cells/6-cm feeder

plate. On day 7, the fibroblast medium was replaced by a human embryonic stem cell medium, which was changed every second day afterward. Around day 20, iPSC growth foci became ready for cutting. The colonies were picked mechanically using a scalpel and plated on freshly gelatinized 4-well dishes with feeder cells and human embryonic stem cell medium.

A few initial passages were done by mechanical cutting with a scalpel, and the cells were kept on feeder cells with human embryonic stem cell medium. Afterward, the cells were transferred on Matrigel-coated plates and cultured further in E8 medium. For passaging, the cells were washed once with phosphate-buffered saline (1xPBS), after which they were incubated with 0.5 mM ethylenediaminetetraacetic acid (Invitrogen, 15575-038) for 5 min at room temperature. The ethylenediaminetetraacetic acid was gently removed, and fresh E8 medium was added to the dish to resuspend the cells, which were distributed to new plates. The same procedure as above was followed for freezing, but the cells were resuspended in ice-cold FBS with 10% dimethyl sulfoxide (Sigma-Aldrich, D8418) and then added to a cryovial cooled on ice. The patient and control iPSCs were expanded over passage 30, with cells frozen as stocks at different passages. The characterization of iPS cells involved the expression of pluripotency markers assessed by immunofluorescence (Supplementary Fig. 7). In this study, one iPSC line derived from Patient 1 and iPSC lines from three control individuals were used.

**Cardiomyocyte differentiation and characterization**. For cardiomyocyte differentiation, we followed the protocol published by Burridge et al.[39], with a few modifications. The iPSCs were grown on Matrigel (Corning, 354277) in E8 media for 3–4 days until they became 65–85% confluent. At this stage, they were dissociated into single cells after incubation with 0.5 mM ethylenediaminetetraacetic acid in 1xPBS for 7 min. The iPSCs were seeded at 120,000 cells/well of a 6-well dish coated with growth factor-reduced Matrigel (Corning, 356230) in the presence of 10 µM Rock inhibitor (Y27632). The E8 medium was regularly changed until cells reached confluence and the differentiation process was started.

For cardiomyocyte induction, CDM3 glucose medium with 3–4.5 µM CHIR99021 (Sigma), a GSK3B inhibitor that mediates epithelial to mesenchymal transition, was used. After 48 h, the cells were washed with 1xPBS, and the medium was changed with CDM3 glucose containing 2 µM Wnt-C59, a WNT inhibitor. After another 48 h, the cells were washed, and the medium was changed with normal CDM3 glucose, which was refreshed every other day from this point. The cells typically started to beat between days 7 and 10. The cells were replated on day 20 at a higher density of 1–1.5 million cells/well of a 24-well plate coated with Matrigel. For cardiomyocyte dissociation, the Multi Tissue Dissociation Kit 3 (Miltenyi Biotech, 130-110-204) was used, and for purification iPSC-Derived Cardiomyocyte Isolation Kit, Human (Miltenyi Biotech, 130-110-188), following manufacturer's instructions. After the cells were passed through the purification kit column, the enriched cardiomyocytes were collected in 15 mL Falcon tubes topped with CDM3 glucose medium and 20% FBS. The cells were pelleted at $500 \times g$ for 5 min or longer, when necessary, resuspended in 1 mL CDM3 glucose containing 20% FBS and 10 µM Rock inhibitor (Y27632), counted and seeded. When the attachment is proper and the cells are healthy, they start to contract after 2–5 days.

At day 30 of culturing in the CDM3 glucose medium, the maturation of cells was started by switching to CDM3 galactose and fatty acids. The maturation continued for 15 days. Around day 45, the cardiomyocytes were dissociated and replated on Matrigel-coated coverslips for immunofluorescence staining. The seeding density was about 125,000 cells/well of a 24-well plate for

immunofluorescence. The cells were fixed 3–4 days after replating. A subset of cells was fixed using a relaxing buffer to obtain distended sarcomeres (fixation conditions detailed in the "Immunocytochemistry" section). Cardiomyocyte cultures from Patient 1 and Control 1 showed >80% purity, while Control 2 and Control 3 showed 30–50%. The purity was estimated by immunofluorescence.

**Immunoblot analysis and antibodies**. Total cellular proteins were extracted in radioimmunoprecipitation assay buffer from cell pellets or tissues stored at −80 °C. Protein concentration was determined with Pierce™ BCA Protein Assay Kit (Thermo Fisher), and equal amounts of total proteins were separated on a 4–20% gradient polyacrylamide gel with stain-free loading control (Mini-PROTEAN TGX Stain-Free™ Gels, Bio-Rad). The separated proteins were transferred on polyvinylidene fluoride membranes, blocked in 5% milk in tris(hydroxymethyl)aminomethane-buffered saline, 0.1% Tween® 20 (1xTBST), and incubated with primary antibody (typically overnight, 1:10,000 dilution), followed by washes with 1xTBST and incubation with secondary antibody for 1 h at room temperature. The chemiluminescent reaction (Clarity™ Western ECL substrate, Bio-Rad) was used to develop the signal, and the image was captured with Bio-Rad ChemiDoc™ XRS+ imager. Primary antibodies were against TMOD1 (1:10,000, Proteintech, 10145-1-AP, host rabbit), TMOD4 (1:10,000, Sigma, SAB1101538, host rabbit), Vinculin (1:10,000, Abcam, ab129002, host rabbit).

**Overexpression in rat cardiomyocytes**. Rat cardiomyocytes were isolated from postnatal day 3 (PD 3) or younger of mixed gender Sprague-Dawley rats. Rat hearts were digested with shaking at 230 RPM at 37 °C in the presence of pancreatin and collagenase 5 times, 15 min each. Isolated cells were plated in 35-mm tissue-culture dishes on Matrigel (BD Biosciences)-coated (1:1000) 12 mm-diameter, number 1.5 coverslips (Fisher Scientific) at ~450,000 cells per dish. Cells were transfected with plasmids expressing GFP, GFP-TMOD1$^{wt}$ or GFP-TMOD1$^{R189W}$ using Lipofectamine™ 3000 Transfection Reagent (Thermo Fisher) ~5 h after plating.

**Immunocytochemistry and antibodies**. iPSCs and cardiomyocytes were pre-fixed for 10 min by adding an equal volume of 4% paraformaldehyde to their medium, followed by 10 min of fixation with 4% paraformaldehyde and three washes with 1xPBS. The fixed cells were kept at 4 °C in 1xPBS until stained. For staining, the cells were first permeabilized for 15 min using 0.2% Triton X-100 (Sigma-Aldrich, X100-1L) in 1xPBS, washed with 1xPBS, followed by a blocking step of 1 h with 10% horse serum (Gibco, 26-050-088), 1% bovine serum albumin (BSA) (Sigma-Aldrich, A4503-50G), and 0.1% Triton X-100 in 1xPBS. The cells were washed once with 1% horse serum, 1% BSA, and 0.1% Triton X-100 in 1xPBS and incubated with primary antibodies overnight at 4 °C. The primary antibodies were diluted in 1% horse serum, 1% BSA, and 0.1% Triton X-100 in 1xPBS. The next day, the cells were washed three times with 1% BSA in 1xPBS and incubated for 3 h with Alexa-Fluor™ 568 Phalloidin diluted in 1% BSA in 1xPBS before one more hour of incubation together with secondary antibodies. The 3-h incubation with phalloidin was done to enhance F-actin staining in iPSC-cardiomyocytes, which otherwise was not strong. The final washes were three times with 1xPBS and one time with phosphate buffer. The coverslips were mounted on glass using Vectashield Mounting Medium with DAPI (Vector Laboratories).

Two to three days after transfection, neonatal rat cardiomyocytes were washed twice with 1xPBS and incubated in relaxing buffer (10 mM 3-(N-morpholino)propanesulfonic acid, pH 7.4, 150 mM potassium chloride, 5 mM magnesium chloride, 1 mM ethylene glycol-bis(β-aminoethyl ether)-N,N,N′,N′-tetraacetic acid, 4 mM adenosine triphosphate) for 15 min and fixed with 2% paraformaldehyde in relaxing buffer for 15 min, at room temperature. Cells were permeabilized in 0.2% Triton X-100 in 1xPBS for 20 min at room temperature and incubated with blocking buffer (2% BSA, 1% normal donkey serum in 1xPBS) for 1 h at room temperature. Cells were incubated with primary antibodies diluted in a blocking buffer overnight at 4 °C. The next day coverslips were washed with 1xTBST 5 times, 5 min each and incubated with Alexa Fluor™ 405-conjugated goat anti-mouse IgG (1:200) and Texas Red™-X 641 Phalloidin (1:50) (Thermo Fisher) diluted in 1xTBST for 2 h at room temperature. Coverslips were then washed with 1xTBST 5 times, 5 min each, and mounted onto glass slides with Aqua Poly/Mount (Polysciences Inc, Warrington, PA) for deconvolution microscopy.

The primary antibodies were: TNNT2: cardiac troponin T (1:300, Abcam, ab45932, host rabbit; and 1:100, Novus Biologicals, MAB1874, host mouse), TMOD1: Tropomodulin 1 (1:100, Novus Biologicals, NBP2-00955, host mouse), SSEA4: stage-specific embryonic antigen 4 (1:200, Thermo Fisher, MC-813-70, host mouse), TRA-1-60 (1:50, Thermo Fisher, MC-813-70, host mouse), Nanog (1:500, Cell Signaling Technologies, 1E6C4, host mouse), α-actinin (1:200, Sigma-Aldrich, EA-53, host mouse).

The tested antibodies were picked based on the manufacturers' recommendations on cross-reactivity and specificity. Known molecular weights of detected proteins were used as an indicator of specificity in immunoblots. For immunostaining of proteins, we compared our findings to published data to validate our results. Secondary antibodies alone were always used to determine the background signal.

**Thin filament length measurements**. Images of neonatal rat cardiomyocytes were captured using a Nikon Eclipse Ti 646 microscope with a 100x NA 1.5 objective and a digital CMOS camera (ORCA647 flash4.0, Hamamatsu Photonics, Shizuoka Prefecture, Japan). 3D deconvolution was performed using NIS offline deconvolution software (Nikon Corporation, Tokyo, Japan). Thin filament lengths (i.e., from pointed end to pointed end, comprised of two bundles of thin filament arrays extending into opposite half-sarcomeres) and sarcomere lengths (the distance between two Z-discs) were calculated from line scan measurements of phalloidin-stained actin filaments using the Distributed Deconvolution (DDecon) plugin for ImageJ[40,41]. Thin filament lengths (Fig. 4j) were calculated by halving the measurement for two bundles of thin filament arrays that extend into opposite half-sarcomeres (magenta bars in Fig. 4i). Thin filament lengths were analyzed between a sarcomere length range of 1.8–2.0 μm. Sarcomere lengths were not different within this range between the myofibrils from cardiomyocytes expressing GFP, GFP-TMOD1$^{wt}$ or GFP-TMOD1$^{R189W}$.

**Pyrene-actin polymerization assay**. Inhibition rates of actin polymerization from the pointed ends by TMOD1 were measured by the change in pyrene-actin fluorescence using an Agilent Cary Eclipse fluorescence spectrometer (Santa Clara, CA). For experiments with a gelsolin-to-actin molar ratio of 1:10 (Fig. 4a, b), 10 μM G-actin was polymerized into F-actin in the presence of 1 μM gelsolin in F-buffer (25 mM imidazole, pH 7.0, 100 mM potassium chloride, 2 mM magnesium chloride, 1 mM ethylene glycol-bis(β-aminoethyl ether)-N,N,N′,N′-tetraacetic acid, 2 mM tris(hydroxymethyl)aminomethane–hydrochloric acid, 0.2 mM calcium chloride, 0.2 mM adenosine triphosphate, and 0.5 mM dithiothreitol) overnight at 4 °C to prepare actin filaments capped

at their barbed ends with gelsolin. These F-actin seeds were diluted to 0.1 μM with 1 μM G-actin (10% pyrene-actin) in F-buffer in the presence of varying concentrations of TMOD1$^{wt}$ or TMOD1$^{R189W}$ and actin polymerization at the pointed ends was monitored by the increase in fluorescence. For experiments with a gelsolin-to-actin molar ratio of 1:100 (Fig. 4c, d and Supplementary Fig. 5A, B), 10 μM G-actin was polymerized into F-actin in the presence of 0.1 μM gelsolin in F-buffer overnight at 4 °C to prepare actin filaments capped at their barbed ends with gelsolin. F-actin seeds were diluted to 0.6 μM with 1 μM G-actin (10% pyrene-actin) in F-buffer in the absence or presence of 0.2 μM TPM1 and in the presence of varying concentrations of TMOD1$^{wt}$ or TMOD1$^{R189W}$. Actin polymerization at the pointed ends was monitored by the increase in fluorescence. Exponential growth curves to maximum were fitted to the polymerization data using SigmaPlot 12.0, and initial elongation rates relative to controls without TMOD1 were calculated as the first derivatives at time zero. The $K_d$ values (the TMOD1 concentration required for 50% maximal capping activity) were calculated from a one-phase exponential decay model in Prism 9 (GraphPad Software, Inc., San Diego, CA):

$$y = y_{min} + y_{max} \cdot e^{-k \cdot x},$$

where $y_{min}$ and $y_{max}$ are the minimal and maximal relative polymerization rates ($R_{exp} / R_{ctrl}$), respectively, and $k$ is the rate constant. A single $K_d$ was obtained by fitting the nonlinear regression model to the data from each individual experiment and the $K_d$ values reported are mean ± standard deviation obtained from 3 or 4 independent experiments.

**Pyrene-actin depolymerization assay**. Inhibition of actin depolymerization rates from the pointed ends by TMOD1 was measured by the change in pyrene-actin fluorescence using an Agilent Cary Eclipse fluorescence spectrometer (Santa Clara, CA). For experiments with a gelsolin-to-actin molar ratio of 1:10 (Fig. 4e, f), 10 μM G-actin (10% pyrene-actin) was polymerized into F-actin in the presence of 1 μM gelsolin in F-buffer overnight at 4 °C to prepare actin filaments capped at their barbed ends with gelsolin. Pyrene-F-actin seeds were diluted to 0.1 μM in F-buffer in the presence of varying concentrations of TMOD1$^{wt}$ or TMOD1$^{R189W}$, and actin depolymerization at the pointed ends was monitored by the decrease in fluorescence. For experiments with a gelsolin-to-actin molar ratio of 1:100 (Fig. 4g, h and Supplementary Fig. 5C, D), 10 μM G-actin (10% pyrene-actin) was polymerized into F-actin in the presence of 0.1 μM gelsolin in F-buffer overnight at 4 °C to prepare actin filaments capped at their barbed ends with gelsolin. Pyrene-F-actin seeds were diluted to 0.6 μM in F-buffer in the absence or presence of 0.2 μM TPM1 and varying concentrations of TMOD1$^{wt}$ or TMOD1$^{R189W}$, and actin depolymerization at the pointed ends was monitored by the decrease in fluorescence. Exponential decay curves were fitted to the polymerization data using SigmaPlot 12.0, and initial rates relative to control (spontaneous actin depolymerization) were calculated as the first derivatives at time zero. The $K_d$ values (the TMOD1 concentration required for 50% maximal capping activity) were calculated from a one-phase exponential decay model in Prism 9 (GraphPad Software, Inc., San Diego, CA):

$$y = y_{min} + y_{max} \cdot e^{-k \cdot x},$$

where $y_{min}$ and $y_{max}$ are the minimal and maximal relative depolymerization rates ($R_{exp} / R_{ctrl}$), respectively, and $k$ is the rate constant. A single $K_d$ was obtained by fitting the nonlinear regression model to the data from each individual experiment and the $K_d$ values reported are mean ± standard deviation obtained from 3 independent experiments.

**Actin depolymerization assay**. The sedimentation assay for actin filament depolymerization from pointed ends was adapted from Lewis et al.[42] with modifications. Here, 5 μM G-actin in the presence of 50 nM gelsolin was polymerized into F-actin in F-buffer for 1 h at room temperature to create actin filaments with their barbed ends capped by gelsolin. 2 μM TPM1 was added to the filaments and incubated for 30 min, at room temperature. Next, filaments were incubated with differing concentrations of TMOD1$^{wt}$ or TMOD1$^{R189W}$ (0, 12.5, 25, 50, 100, 250 nM) for 30 min, at room temperature. Depolymerization was initiated by diluting actin filaments 5-fold with F-buffer in the presence of 5 μM Lat A (Enzo Life Sciences, Farmingdale, NY). Final concentrations after dilution were 1 μM F-actin; 10 nM gelsolin; 0.4 μM TPM1; 0, 2.5, 5, 10, 20 or 50 nM TMOD1. After incubating for 90 min, at room temperature, the filaments were pelleted at 100,000 RPM (Beckman-Coulter TLA-100 rotor), for 30 min, at 4 °C. The G-actin in the supernatants and F-actin in the pellets were separated and solubilized in 1x Laemmli sample buffer for 30 min, at room temperature. The solubilized samples were boiled at 100 °C, 5 min and resolved on 10% sodium dodecyl sulfate–polyacrylamide gels. Gels were stained with Coomassie Brilliant Blue R-250 (Bio-Rad) and images were captured with an Epson Perfection 2450 Scanner.

Quantitative analysis of the background-corrected band densities of F-actin in the pellet and G-actin in the supernatant for samples was performed using ImageJ (NIH). The band density of actin in each sample was divided by the band density of actin in the absence of Lat A and TMOD1 (control) to calculate relative actin band densities. The $IC_{50}$ values (the concentration required for 50% maximal capping activity by TMOD1) were calculated from a one-phase exponential inhibitor-dose-response model in Prism 9 (GraphPad Software, Inc., San Diego, CA):

$$y = y_{max} + \frac{y_{max} - y_{min}}{1 + 10^{\left(\log\left(IC_{50}-x\right) \cdot n\right)}},$$

where $y_{max}$ and $y_{min}$ are the maximal and minimal relative actin band densities, respectively, $IC_{50}$ is the TMOD1 concentration that prevented 50% of the total filaments from depolymerizing and $n$ is the Hill coefficient. A single $IC_{50}$ value was obtained by fitting the nonlinear regression model to the data from each individual experiment and the $IC_{50}$ values reported are mean ± standard deviation obtained from 3 or 4 independent experiments.

**Statistics and reproducibility**. All statistical analyses were performed in Prism 9 (GraphPad Software, Inc., San Diego, CA). Two-tailed Student's $t$-test or one-way ANOVA with Tukey's post-hoc test was used depending on the number of groups or variables. Differences with $p < 0.05$ were considered statistically significant.

For the cell culture and recombinant protein studies, we initially aimed to perform 3 independent experiments in this manuscript. If data from a certain group(s) failed to reach to a sample size of 3 due to an inability of quantification resulting from technical difficulties (imperfections in gel/staining/culturing), the experiment was repeated for a fourth time in order to reach to a sample size of at least 3 for all groups and to achieve statistical significance. No data were excluded from analysis and all attempts of replication were successful.

For randomization, control cells (GFP-expressing or control iPSC-cardiomyocytes) were grouped and compared to the variant-expressing cells. For thin filament length measurements, folders containing data sets were blinded and revealed after analysis. For certain immunofluorescence sets, blinding was not possible since the treatment (i.e., presence of GFP fluorescence) had to be determined prior to collecting data.

**Data collection**. Chemiluminescence images of immunoblots were captured with Image Lab Software (Bio-Rad). ImageJ Version 1.52 (NIH) was used for collecting data from Coomassie-stained SDS-PAGE gels. ImageJ version 1.52 DDecon plug-in was used for thin filament length measurements. 3D deconvolution was performed using NIS offline deconvolution software (Nikon). Structural modeling of the TMOD1 homozygous variant was performed using Discovery Studio 4.5 (Biovia).

**Data analysis**. GraphPad Prism 9.0 was used for compiling the data, creating the figures and statistical analysis.

**Reporting summary**. Further information on research design is available in the Nature Portfolio Reporting Summary linked to this article.

## Data availability

The raw genetic data are not publicly available to preserve individuals' privacy under the European Data Protection Regulation. Data from the main figures are available in Supplementary Data 1 and Supplementary Fig. 8. All other data are available on request.

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

## Acknowledgements

The authors thank Markus Innilä and Rachel Mayfield for technical support. Biomedicum Imaging Unit facility is acknowledged for providing infrastructure and services. CSC-IT Center for Science Finland is acknowledged for computational resources. The funding was provided by the Finnish Foundation for Cardiovascular Research, Jane and Aatos Erkko Foundation, Sigrid Jusélius Foundation, Academy of Finland, University of

Helsinki, Finnish Cultural Foundation, Alfred Kordelin Foundation, Otto Malm Foundation, Maud Kuistila Foundation, Foundation for Pediatric Research Finland, National Institutes of Health (R01HL123078, R01HL164644), American Heart Association (19POST34450023) and Czarina M. and Humberto S. Lopez Endowed Chair for Excellence in Cardiovascular Research.

## Author contributions

Conceptualization: A.S., T.H.O., M.C., C.V., C.C.G., V.M.F.; Data curation: V.B., S.O., C.J.C., J.K.; Formal analysis: C.V., M.C., C.J.C.; Funding acquisition: A.S., C.J.C., C.V., T.H.O., M.C., C.C.G.; Investigation: C.V., M.C., T.H.O., A.M., K.Y., O.R., T.P., L.M., R.K., H.H., V.B., J.K., T.M.; Methodology: C.V., M.C., T.H.O., R.K., C.C.G., V.M.F., T.M., A.S.; Resources: V.B., S.O., P.L., A.S.; Visualization: C.V., M.C., A.M.; Writing—original draft: C.V., M.C., T.H.O., A.S.; Writing—review & editing: all authors.

## Competing interests

The authors declare the following competing interests: J.K. is co-founder and director of Blueprint Genetics, which offers genetic diagnostics for inherited disorders. The other authors declare no competing interests.

## Ethics approval

The study plan of the Childhood Cardiomyopathy project was approved by the Child and Adolescent Psychiatry Ethical Board and Coordinating Ethical Board of Helsinki University Hospital and received the ethical permit numbers 291/13/03/03/2008 and 254/13/03/00/14. All the samples were taken for diagnostic purposes with informed consent from parents and from patients when they were older than 10 years of age. All ethical regulations relevant to human research participants were followed. The work with animals was performed under the approval by The Institutional Animal Care and Use Committee at the University of Arizona, Protocol number 08-017, which conformed to all applicable federal and institutional policies, procedures and regulations, including the Public Health Service Policy on Humane Care and Use of Laboratory Animals, United States Department of Agriculture regulations (9 CFR Parts 1, 2, 3), the Federal Animal Welfare Act (7 USC 2131 et. Seq.), the Guide for the Care and Use of Laboratory Animals, and all relevant institutional regulations and policies regarding animal care and use at the University of Arizona.
