## [Peer Review File · Communications Biology]

Reviewers' comments:

Reviewer #1 (Remarks to the Author):

Vasilescu et al. have submitted a well written account of their discovery that a missense mutation in tropomodulin 1 results in a type of cardiomyopathy in humans. The conclusions are well-justified from the data presented and are novel. The report will be of interest to cardiologists since most cases of cardiomyopathy are genetic in origin, and yet, when a patient with cardiomyopathy is investigated causative mutations can only be found in about half of the patients. Thus, there is a need to identify new cardiomyopathy genes. Tropomodulins are well-established to cap and thus inhibit actin monomer addition and removal at the pointed end of F-actin, and acting together with tropomyosin and leiomodins, are involved in determining and maintaining the uniform lengths of thin filaments in striated muscle (cardiac and skeletal muscle). Most cardiomyopathy genes encode sarcomeric proteins including those of the thin filament; however, this would be the first report of mutations in the gene specifying tropomodulin 1.

The authors provide convincing data with adequate statistics for the following: (1) Three children with dilated cardiomyopathy from 2 unrelated families are recessive for the R189W mutation. (2) R189 is an absolutely conserved residue in tropomodulins in many animal species. (3) R189 resides in the N-terminal part of the 2nd actin binding region of tmod1 and by placing on a known crystal structure is predicted to affect local folding and/or binding to F-actin. (4) iPSC-derived cardiomyocytes from a patient were obtained and it shows normal sarcomere organization and normal localization of the mutant TMOD1 to the thin filament. (5) Taking advantage of the fact that TMOD1 is also expressed in skeletal muscle, the authors conduct proteomic analysis from one of the patients, and this shows altered levels of various actin binding proteins. (To me, this is the least interesting and informative experiment--see below). (6) In in vitro actin polymerization assays, recombinant R189W TMOD1 inhibits pointed end polymerization less well than wild type TMOD1. (7) In latrunculin A-induced F-actin depolymerization assays, the mutant TMOD1 is less able to inhibit pointed end depolymerization. Thus, both of these assays, demonstrate convincingly that R189W tmod1 has lower than wild type pointed end capping activity. (8) The authors also overexpress GFP tagged wild type and mutant TMOD1 in cardiomyocytes to determine if such overexpression of TMOD1 would reduce the length of thin filaments. Again, the mutant TMOD1 shows less activity--thin filaments are longer in the presence of excess mutant TMOD1 than they are in the presence of wild type TMOD1.

Overall, I believe the authors have convinced us that the R189W recessive mutation tmod1 reduces the normal activity of tmod and that it is responsible for development of dilated cardiomyopathy. Thus, I would recommend publication if the authors can address the following comments or suggestions.

1. The authors indicate in the Discussion that the patients do not develop skeletal myopathy. And yet, they perform proteomic analysis of skeletal muscle from one of the patients. I question whether this analysis is relevant to understanding the cardiomyopathy.
2. A prominent early feature of the cardiomyopathy in all 3 patients is the atrial dilation without ventricular dilation. Could the authors talk about this in the Discussion? Is it possible that the reason dysfunction first occurs in atria is that in atria as compared to ventricles, the level of TMOD1 is lower, or that the level TMOD1 is the same but the level of TMOD3 or leiomodin is lower?
3. It is also puzzling as to why it takes 12-14 years for the cardiomyopathy to become manifest. Of course, this is a fundamental question for the development of most cardiomyopathies, especially those of adult onset. Their data shows that the R189W TMOD1 has reduced but not absent activity. Is it

possible that the reason for the delay in displaying the phenotype is that TMOD1 is expressed at high level in the fetus, baby and young child, but gradually declines with age so that a threshold is breached in which the combination of lower expression and the lower activity of the mutant TMOD1 is enough to show the phenotype?

4. The only disappointment is that they have not shown us that the length of thin filaments are increased in the patients. Without this, how can the authors explain the development of cardiomyopathy? Perhaps this is asking for too much, but couldn't they perform fluorescence microscopy (perhaps even super resolution if necessary) on patient heart tissue using phalloidin and a marker for the M-line (e.g. myomesin or the C-terminus of titin). If the mutant TMOD1 really has reduced activity, they should observe an extension of the actin filaments into the M-line that would not occur in normal samples. Two of the patients underwent cardiac transplantation so their hearts were collected when they were having maximum signs and symptoms. Alternatively, if they think that the skeletal muscle biopsies are indeed relevant, perhaps they can do a similar analysis of the patient's skeletal muscle biopsies.

Reviewer #2 (Remarks to the Author):

The work by Catalina Vasilescu and coauthors reports a mutation in an actin pointed-end capping protein Tmod1 as a source of inherited pediatric cardiomyopathy. The same Tmod1 mutation was detected in three cardiomyopathy patients, two of which are siblings. Magnetic resonance imaging revealed dilated atria with consistent diastolic dysfunction and various degrees of systolic dysfunction. Biochemical analysis of the recombinant Tmod1 R189W variant revealed an affected ability of the protein to inhibit polymerization at the pointed end, which correlated with elongated thin filaments in cardiomyocytes in culture expressing the mutated protein. Mass spectrometry analysis of a patient's skeletal muscle revealed normal levels of Tmod1 and Tmod4 but altered levels of other proteins, e.g., related to the actin cytoskeleton, involved in energy metabolism and protein homeostasis. Overall, this work is the first report that mutations in Tmod1 are involved in pediatric cardiomyopathies. The work is conducted at a sound technical level and is well written. That being said, I have a few concerns and suggestions listed below.

Concerns:

1. Fig 4A shows a measurably less effective inhibition of filament elongation at their pointed end, which is also reflected in a less effective inhibition shown in Fig 4B. My concern is that the inhibition by wt Tmod1 is several-fold less effective than reported previously (PMID:28494946; 25061212). Can the authors speculate about the source of this difference?
2. For the same figure, given that tropomyosin (TM) is an essential contributor to Tmod's function under physiological conditions, including TM in this experiment is desirable.
3. Fig 4C-D. It is not clear why a less informative and potentially less accurate end-point assay (pelleting) was chosen to demonstrate the depolymerization in the presence of Tmod1 instead of continuously monitoring filament depolymerization by the pyrene fluorescence assay. Supplementing the experiment with pyrene actin depolymerization would strengthen the conclusions. Also, showing supernatants for the pelleting assay (perhaps in the supplement) in addition to pellets is an important internal control that should be included.
4. Fig 4E-F. Given that the shortening of sarcomeres is Tmod1 concentration dependent, the expression levels of wtTmod1 vs. mutantTmod1 should be evaluated to strengthen the conclusion.

Minor suggestions:

1. Fig. 2F. To my knowledge, the skeletal muscle does not perform gluconeogenesis. The reason

gluconeogenesis was highlighted by the software is that it shares numerous reactions with glycolysis, which is certainly active in the muscle.

2. Why did phalloidin stain F-actin only in/near the Z-discs and not through the entire thin filament length? Are the authors aware of any competition with ABDs that can account for this phenomenon? Please provide references for published works with similar staining patterns.

3. Speculating why the atrial and not ventricular functions are the first to suffer in Tmd1-dependent cardiomyopathy would be desirable.

Reviewer #3 (Remarks to the Author):

This paper is the first to give an insight into the role of TMOD1 in human childhood cardiomyopathy. This is really exciting and the characterisation data provide a plausible explanation why the TMOD1 R189W mutant may not be able to do its job properly. The data are very convincing and well presented and I have just a few minor comments:

1. Figure 4E&F: The quantification of thin filament length is unclear at the moment. The phalloidin obviously labels the entire I-band; as does the magenta line that is drawn, but the measurements are too short. Was the value for the magenta line then halved to come up with the values in F? If so, please state in Material & Methods.

2. In Figure 3 it almost appears as if the signal for the Tmod 1 in the patient cells is a bit broader, sometimes almost in doublets, suggesting actually shorter thin filaments? Would not really fit with the biochemical data, but can the authors please go back to other images from the patient cells to check whether the image is representative?

3. There is a recent paper in PNAS on a human FLII gene variant that causes cardiomyopathy and may do so via its interaction with Tmod1 (Kuwabara et al.; PNAS 120; 2023) - please discuss and cite.

We would like to thank the Reviewers for their valuable input, which allowed us to improve our study and clarify the representation of our data. The Reviewers had many positive comments including: “The conclusions are well-justified from the data presented and are novel.”, “The report will be of interest to cardiologists...”, “The authors provide convincing data with adequate statistics...”, “The work is conducted at a sound technical level and is well written” and “The data are very convincing and well presented”. The changes in the revised manuscript are highlighted in yellow. Following the reviewers’ suggestions, we significantly expanded Figure 4 and added the following new tables and figures: Supplementary Table 1, 2, 3 and Supplementary Figures 6, 7. Our point-by-point responses to the Reviewers’ comments are listed below.

Reviewer #1

1. The authors indicate in the Discussion that the patients do not develop skeletal myopathy. And yet, they perform proteomic analysis of skeletal muscle from one of the patients. I question whether this analysis is relevant to understanding the cardiomyopathy.

This is a good point. Because of this comment and low n-number, we have now omitted the proteomics data. Unfortunately, we did not have cardiac tissue available from the patient.

2. A prominent early feature of the cardiomyopathy in all 3 patients is the atrial dilation without ventricular dilation. Could the authors talk about this in the Discussion? Is it possible that the reason dysfunction first occurs in atria is that in atria as compared to ventricles, the level of TMOD1 is lower, or that the level TMOD1 is the same but the level of TMOD3 or leiomodoin is lower?

Atrial dilatation is a consequence of poor ventricular function. Restrictive cardiomyopathy (RCM) is a form of cardiomyopathy in which the ventricular walls of the heart are rigid (restrictive). Thus, the ventricles are restricted from stretching and filling with blood properly, leading secondarily to an increase in atrial filling and atrial enlargement. Therefore, the size of the atria reflects the degree of diastolic dysfunction in the ventricles. We have added a short discussion of this, as suggested by the Reviewer (page 6, lines 45-50).

3. It is also puzzling as to why it takes 12-14 years for the cardiomyopathy to become manifest. Of course, this is a fundamental question for the development of most cardiomyopathies, especially those of adult onset. Their data shows that the R189W TMOD1 has reduced but not absent activity. Is it possible that the reason for the delay in displaying the phenotype is that TMOD1 is expressed at high level in the fetus, baby and young child, but gradually declines with age so that a threshold is breached in which the combination of lower expression and the lower activity of the mutant TMOD1 is enough to show the phenotype?

Thank you for this thoughtful discussion. Expression changes in human postnatal heart development are a possibility, but as the reviewer states, it is the same question for all cardiomyopathies - why do they manifest at a certain age? We are not aware of data indicating changes in the expression of TMOD1 during the human lifespan, but it is a topic worth exploring in the future, when suitable human tissue becomes available.

4. The only disappointment is that they have not shown us that the length of thin filaments are increased in the patients. Without this, how can the authors explain the development of cardiomyopathy? Perhaps this is asking for too much, but couldn't they perform fluorescence microscopy (perhaps even super resolution if necessary) on patient heart tissue using phalloidin and a marker for the M-line (e.g. myomesin or the C-terminus of titin). If the mutant TMOD1 really has reduced activity, they should observe an extension of the actin filaments into the M-line that

would not occur in normal samples. Two of the patients underwent cardiac transplantation so their hearts were collected when they were having maximum signs and symptoms. Alternatively, if they think that the skeletal muscle biopsies are indeed relevant, perhaps they can do a similar analysis of the patient's skeletal muscle biopsies.

We agree that this would be interesting, but we do not have suitable tissue to provide relevant data. The terminal stage of the RCM heart after transplantation with fibrotic, rigid tissue was too challenging to interpret; accurate analysis should be performed on sarcomeres that are relaxed, with patient and control materials treated similarly. Note, even in the optimal conditions of a cell culture, we showed a small difference in actin filament length. We omitted the skeletal muscle data, as explained above (Point #1).

Reviewer #2

1. Fig 4A shows a measurably less effective inhibition of filament elongation at their pointed end, which is also reflected in a less effective inhibition shown in Fig 4B. My concern is that the inhibition by wt Tmod1 is several-fold less effective than reported previously (PMID:28494946; 25061212). Can the authors speculate about the source of this difference?

We initially used a 1:100 gelsolin-to-actin molar ratio to form the filaments for this experiment. Spontaneous fragmentation of filaments when not stabilized by TPM1 can reveal new barbed ends that are not capped due to insufficient concentration of the barbed-end capping protein (gelsolin). We hypothesize that this phenomenon may have created a background level of barbed end elongation and the subsequent fluorescence increase from barbed end elongation may have affected the measurement for TMOD1's pointed end-capping activity. We repeated these experiments with shorter filaments at a 1:10 gelsolin-to-actin molar ratio and now show the new results in Fig. 4, Supplemental Fig. 6 and Supplemental Table 1 and 2. The affinity measurements we now report are well within range of the K_d values for TMOD1 that were published previously. However, please note that Rao et al. 2014 used CapZ as the barbed-end capping protein for their assays (while we used gelsolin); this difference may be a contributing factor for the differences observed in K_d measurements between these two studies.

2. For the same figure, given that tropomyosin (TM) is an essential contributor to Tmod's function under physiological conditions, including TM in this experiment is desirable.

We updated Fig. 4 to include panels C, D, G and H, which now demonstrate the effect of tropomyosin on the capping activity of TMOD1^{wt} and TMOD1^{R189W}. These experiments show that the R189W mutation also reduces the pointed end capping activity of TMOD1 for tropomyosin-coated actin filaments.

3. Fig 4C-D. It is not clear why a less informative and potentially less accurate end-point assay (pelleting) was chosen to demonstrate the depolymerization in the presence of Tmod1 instead of continuously monitoring filament depolymerization by the pyrene fluorescence assay. Supplementing the experiment with pyrene actin depolymerization would strengthen the conclusions. Also, showing supernatants for the pelletting assay (perhaps in the supplement) in addition to pellets is an important internal control that should be included.

We initially included the sedimentation experiment as an independent method to measure the capping activity of TMOD1. We agree with the Reviewer that this is a less sensitive method compared to fluorometry. We now demonstrate the ability of TMOD1 to inhibit pointed end depolymerization of actin filaments using pyrene-actin assays in Fig. 4E-H and moved the sedimentation assay to the Supplementary Information (Supplemental Table 3 and Supplemental Fig. 7).

4. Fig 4E-F. Given that the shortening of sarcomeres is Tmod1 concentration dependent, the expression levels of wtTmod1 vs. mutantTmod1 should be evaluated to strengthen the conclusion.

Please note, we restricted our thin filament length measurements to cells that contained myofibrils with a sarcomere length of 1.8 μm to 2.0 μm (see page 11, lines 11-12) to normalize our measurements between the three different groups (GFP, GFP-TMOD1^{wt}, GFP-TMOD1^{R189W}). As a result, these cells are expected to express comparable levels of GFP-TMOD1^{wt} and GFP-TMOD1^{R189W}; thus, the observed differences in thin filament lengths cannot be caused by shortening of sarcomeres unevenly between the measured groups.

Minor suggestions:

1. Fig. 2F. To my knowledge, the skeletal muscle does not perform gluconeogenesis. The reason gluconeogenesis was highlighted by the software is that it shares numerous reactions with glycolysis, which is certainly active in the muscle.

We have now omitted the proteomics data (See Reviewer #1, Point #1).

2. Why did phalloidin stain F-actin only in/near the Z-discs and not through the entire thin filament length? Are the authors aware of any competition with ABDs that can account for this phenomenon? Please provide references for published works with similar staining patterns.

The preferential phalloidin staining near/in the Z disk area is observed in cardiac myocytes in which sarcomeres are not fully relaxed. We have now added a reference (Zhukarev et al. 1997) that explains this phenomenon (page 4, line 52).

3. Speculating why the atrial and not ventricular functions are the first to suffer in Tmd1-dependent cardiomyopathy would be desirable.

We added to the Discussion the following: "The atrial dilatation that is present in the patients with the TMOD1 p.R189W missense variant is a consequence of poor ventricular function. In restrictive cardiomyopathy (RCM) the ventricular walls of the heart are rigid and cannot expand to fill up with blood sufficiently. This leads secondarily to an increase in atrial filling and dilatation. Therefore, RCM is primarily a disorder affecting the ventricular muscle and the increased atria size reflects the degree of diastolic dysfunction in the ventricles." (page 6, lines 45-50).

Reviewer #3

1. Figure 4E&F: The quantification of thin filament length is unclear at the moment. The phalloidin obviously labels the entire I-band; as does the magenta line that is drawn, but the measurements too short. Was the value for the magenta line then halved to come up with the values in F? If so, please state in Material & Methods.

That is correct. Thank you for bringing it to our attention. We now explain this in Material & Methods: "Thin filament lengths (Fig. 4J) were calculated by halving the measurement for two bundles of thin filament arrays that extend into opposite half-sarcomeres (magenta bars in Fig. 4I)" (page 11, lines 9-11).

2. In Figure 3 it almost appears as if the signal for the Tmod 1 in the patient cells is a bit broader, sometimes almost in doublets, suggesting actually shorter thin filaments? Would not really fit with the biochemical data, but can the authors please go back to other images from the patient cells to check whether the image is representative?

This appearance of Tmod in both control and patient samples as a broader band (doublet) is sporadic and usually linked to the precise relaxation of the sarcomere and/or the plane of the myofibril that is being photographed. We now note this finding in the figure legend.

3. There is a recent paper in PNAS on a human FLII gene variant that causes cardiomyopathy and may do so via its interaction with Tmod1 (Kuwabara et al.; PNAS 120; 2023) - please discuss and cite.

Thank you for this observation. We now cite this paper in the Discussion: “Recently, the human pathogenic p.R1243H mutation in the Flightless-I homolog (FLII) gene, predisposing to cardiomyopathy, was investigated in a series of mouse models. These murine studies suggest that FLII gene may regulate actin thin filament length by interacting with TMOD1 and preventing it from capping pointed ends. (Kuwabara et al., 2023)” (page 6, lines 24-27).

REVIEWERS' COMMENTS:

Reviewer #1 (Remarks to the Author):

The authors have carefully considered my previous comments and suggestions and the paper is now improved and ready for publication.

Reviewer #2 (Remarks to the Author):

The authors addressed all my concerns.

Reviewer #3 (Remarks to the Author):

All my questions were answered to my satisfaction by the authors.

We would like to thank the Reviewers for their careful evaluation of our manuscript. The Reviewers had no additional requests for revision per their comments provided below. The only changes that we made on the manuscript were related to editorial formatting requirements.

Reviewer #1

The authors have carefully considered my previous comments and suggestions and the paper is now improved and ready for publication.

Reviewer #2

The authors addressed all my concerns.

Reviewer #3

All my questions were answered to my satisfaction by the authors.